# EEEC: EMOTION-EXPERIENCER-EVENT-CAUSE MULTI-STEP CHAIN REASONING FOR EMOTION-CAUSE PAIR EXTRACTION

## ABSTRACT

Emotion-cause pair extraction (ECPE) aims to identify all emotion and cause clauses in documents, forming the ECPs. Although existing methods have achieved some success, they face issues such as overlooking the impact of emotion experiencers, failing to leverage specific domain knowledge, and tending to spurious correlations. To address these issues, we transform the ECPE task into a multi-step reasoning problem and propose the Emotion-Experience-Event-Cause (EEEC) framework. We introduce an experiencer identification task to understand the source of emotions and enhance the association between emotion and cause clauses. In addition, by combining both prior knowledge and induced reasoning, EEEC guides a large-scale language model (LLM) to perform the emotion-reason pair extraction task efficiently. Experimental results demonstrate that EEEC achieves performance close to current state-of-the-art supervised fine-tuning methods. The data and code are released at https://anonymous.4open.science/r/EEEC-EB80/

## 1 INTRODUCTION

With the proliferation of social media data, sentiment analysis has emerged as a highly regarded research area. Emotion Cause Analysis (ECA), as a significant branch, is devoted to exploring deeply the underlying motivations and causes when individuals express emotions. By thoroughly examining the logic and factors influencing emotions, it can enhance the capability of intelligent systems to comprehend and respond to human emotions.

Depending on the task's specific setting, ECA tasks can be split into Emotion-Cause Extraction (ECE) and Emotion-Cause Pair Extraction (ECPE). The objective of the ECE task is to extract the cause underlying the emotion in a given document with the expression of the emotion Lee et al. (2010); Gui et al. (2016; 2017); Li et al. (2018); Xia et al. (2019); Ding et al. (2019). ECPE aims to simultaneously extract all the potential emotions and the associated causes from a given document Xia & Ding (2019). However, the former needs to manually annotate the emotion, which not only consumes a lot of human resources but also limits its wide application in real-world scenarios. We focus on the ECPE task, as shown in Figure 1, the document has 18th clauses, and C7 shows the emotion "sadness" caused by C6. C18 expresses a feeling of "sadness," which relates to C17. Therefore, the goal is to extract all emotion cause pairs (ECPs):{(C7, C6), (C18, C17)}.

The essence of the ECPE task hinges on uncovering the relationship between emotion and cause clauses. Intuitively, the specific position relationship between clauses is the distinctive feature for mining the causal relationship between emotion clauses and cause clauses. Consequently, much of the research directly utilizes the relative positions between clauses as features for ECPE task, such as: Ding et al. (2020b); Fan et al. (2020b); Yuan et al. (2020); Cheng et al. (2020); Ding et al. (2020a); Fan et al. (2020a); Wei et al. (2020); Sun et al. (2021); Fan et al. (2021); Singh et al. (2021); Huang et al. (2023); Li et al. (2024). Nevertheless, this approach is only applicable to data that is sensitive to positional features, and the method has poor generalization ability.

Except relying on the prior rules, some studies employ the correlations between clauses Xia & Ding (2019); Wu et al. (2020); Chen et al. (2022b;a;b); Feng et al. (2023); Chen & Mao (2023); Li et al. (2023a); Chen et al. (2020b); Liu et al. (2022) to achieve ECPE. In addition, other studies

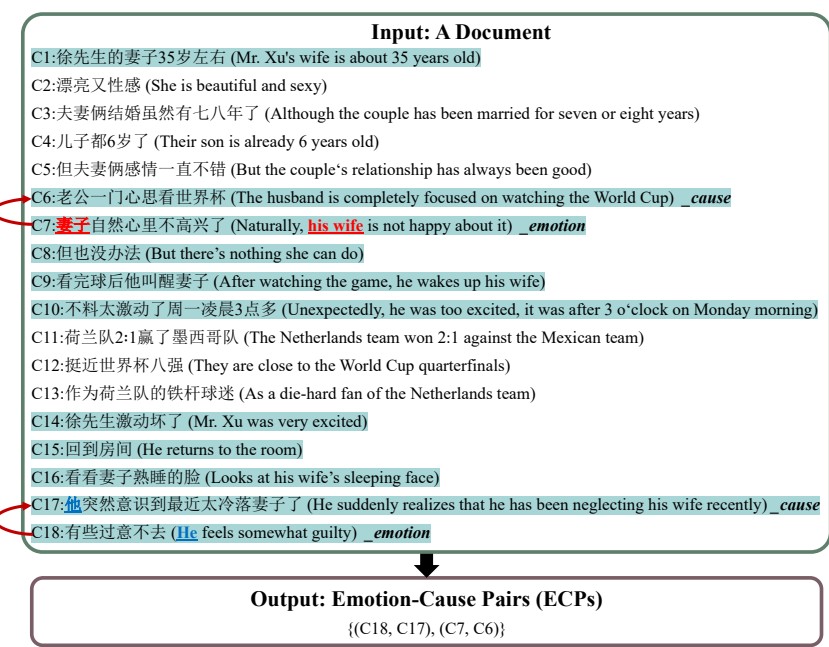

Figure 1: An example of the ECPE task. Experiencers are underlined and highlighted, and different colors are used to distinguish different experiencers; _emotion and _cause refer to the emotion and cause clauses, respectively, and the arrows indicate the cause clauses corresponding to the emotion. In addition, the green highlighted parts represent clauses related to the experiencer.

attempted to integrate emotion type labels Chen et al. (2020a); Tang et al. (2020); Song et al. (2023), emotion keywords Bao et al. (2022a), explicit semantic information of clauses Zhou et al. (2022), commonsense Li et al. (2023b); Gu et al. (2024) to handle the ECPE task. However, these supervised learning tasks require large amounts of labelled data for training.

Recently, the rise of large language models (LLMs), such as ChatGPT [1] and LLaMA Touvron et al. (2023), has demonstrated remarkable performance in multiple NLP tasks under zero-shot or few-shot scenarios, even without updating parameters. Given their powerful semantic understanding and reasoning capabilities, some studies Wang et al. (2023); Wu et al. (2024) have attempted to utilize LLMs to implement ECPE tasks.

Although these approaches have achieved good performance, they still have several shortcomings. (1) **Ignore the influence of the experiencer on the ECPE task**. Experiencers are the person or sentient entity who experiences or feels the emotions Ghazi et al. (2015). (2) **Neglect of domain-specific knowledge**. Emotional clauses, in general, are characterized by specific cue words in the clauses that can serve as a priori domain knowledge for the model. (3) **Exist the spurious correlations**. The existing approaches mainly concentrate on modeling the whole document, which will inevitably consider redundant information, leading to spurious correlations. As shown in Figure 1, the document has two experiencers: "Mr.Xu" and "His wife". "His wife" is the experiencer for C7, while "Mr.Xu" is the experiencer for C18 and C17. Both experiencers correspond to the emotion type "sadness." Given the specific definition of the experiencer, we know that each experiencer may express one or more emotions. Additionally, it is worth noting that experiencers are directly associated with emotion and cause clauses, thus determining the practical scope of emotional and causal clauses. In this case, C2, C3, C4, C5, C11, C12 and C13 will not be candidates for emotion and cause clauses.

In order to tackle these issues, we crafted an Emotion-Experiencer-Event-Cause multi-step chain (EEEC) framework, which can incorporate prior domain knowledge and leverage the association between experiencers and clauses. The framework aims to guide the model, focusing on task-relevant details and mitigating other information interference. In specifically, we decompose the ECPE task

---

[1]https://chat.openai.com/

into five parts: Knowledge-Guided Emotion Recognition, Emotion Classification & Experiencer Recognition, Event Extraction, Analysis, and Validate. Each step involves a prompt interacting with a LLM, and the output of each step serves as part of the prompt for the next step. The first step incorporates prompts that fuse prior knowledge from the emotion analysis domain. And the entire framework operates in a pipeline manner.

Overall, our contributions can be summarized as follows:

- We propose an Emotion-Experience-Event-Cause (EEEC) framework by decomposing the ECPE task into a multi-step reasoning problem. Each step is an elaborated sub-problem for the task.
- To reduce the cascade error generated by emotion extraction for the ECPE task, we introduced prior knowledge from the domain of sentiment analysis in the prompt template to improve the accuracy of the emotion extraction sub-task.
- We emphasize and integrate the critical step of experiencer identification, which allows the framework to more accurately parse and understand the sources and impacts of sentiment, contributing to the filtering of candidate cause clauses.

## 2 RELATED WORK

Xia & Ding (2019) initially presented the Emotion-Cause Pair Extraction (ECPE) task, which aims to extract emotion clauses (EC) and cause clauses (CC) from unannotated documents simultaneously. They designed a **2-step pipeline framework** that can extract EE and CE sequentially. However, it is prone to cascading errors. Subsequently, a considerable number of end-to-end methods have emerged. These **end-to-end methods** can be classified into: method for enumerating and filtering all clause pairs Chen et al. (2020b); Ding et al. (2020b); Wei et al. (2020); Chen et al. (2022b); Wu et al. (2022); Hu et al. (2023). For example, Chen et al. (2020b) construct a relationship graph of all candidate clause pairs and encode the candidate pairs using a graph model to extract ECPs. Ding et al. (2020b) propose an emotion/cause-oriented sliding window mechanism for filtering the candidate clauses. Wei et al. (2020) consider all candidate clause pairs and then perform the ECPE in a ranking manner. Chen et al. (2022b) employs KL divergence to constrain the candidate pair score matrix and align multi-task features to extract ECPs. Wu et al. (2022) transform the ECPE task as a clause-pairs tagging task for any two clause pairs in a document. Hu et al. (2023) treat all clauses as filtering targets and extract the ECPs using genuine and fake pair supervision mechanisms. Sequence labeling was another popular ECPE method, pairing clauses according to elaborate labels. Yuan et al. (2020) construct a labeling scheme based on the distances between emotion clauses and cause clauses to achieve end-to-end ECPE. Chen et al. (2020a) encode the sentiment categories and clause types into tags so that ECPs of different emotion types can be easily distinguished.

Moreover, some researchers introduced additional knowledge for modelling the context of ECPE tasks. Several researchers noticed that explicit semantic information or question answering (QA)-based queries could improve the performance of ECPE tasks. For example, Nguyen & Nguyen (2023) cast the Emotion-Cause Pair Extraction task to the question answering problem. Chang et al. (2022) proposed a two-stage MRC framework through the questing-answering formulation. Zhou et al. (2022) design a Multi-turn MRC framework (MM-R) with Rethink mechanism. Cheng et al. (2023) propose a dual-MRC framework to extract ECPs in a dual-direction way. This framework extracts ECPs by querying two directions and then uses a combination strategy to merge the results obtained from the two directions. Others are devoted to introducing additional knowledge for modelling the context of ECPE tasks. Turcan et al. (2021) achieves emotion classification and emotion cause tagging by combining common sense knowledge with multi-task learning through adaptive knowledge modelling. Bao et al. (2022b) perceive textual semantics through fine-grained semantics introduced by keywords and coarse-grained semantic features between clauses. Zheng et al. (2022) proposed a prompt learning based Emotion-Cause Pair Extraction approach. This method transforms the three sub-tasks of ECPE into three sub-objectives and realizes the extraction of multiple sub-tasks by constructing special prompts. Chen et al. (2022b) aligned features and tasks to model interactions between specific emotion-cause features and tasks explicitly. Chen et al. (2023) proposed a novel two-stage model to handle the ECPE task and incorporate reinforcement learning (RL) to tackle the cascading error issue. Li et al. (2023b) proposed an Experiencer-Driven and Knowledge-Aware Graph Model (EDKA-GM) to enhance the impact of experiencer features

in ECPE tasks. They leveraged ATOMIC's semantic correlations to construct inter-clause relationships, enriching the task's contextual understanding. Gu et al. (2024) suggested an Emotion knowledge-aware Prompt-tuning for Emotion-Cause Pair Extraction (EmoPrompt-ECPE), which achieves the integration of the three subtasks of ECPE by utilizing additional sentiment knowledge and mining the implicit knowledge of cause clauses. Zong et al. (2024) categorize clauses into four distinct types and introduce a knowledge-based multi-classification subtask.

Unlike the above approaches, Wang et al. (2023) pioneered the exploration of the ECPE task in a zero-shot scenario, enabling the model to identify one of the most evident ECPs through multi-step reasoning. Additionally, Wu et al. (2024) custom-designed a Decomposition of Emotional Cause Chaining (DECC) framework, which decomposes the ECPE task into four steps: recognizing, locating, analyzing, and summarizing. However, the existing ChatGPT-based methods have yet to yield satisfactory results. Therefore, we aim to enhance the chain-of-thought framework by incorporating additional knowledge.

## 3 APPROACH

### 3.1 MOTIVATION

For the benchmark dataset Xia & Ding (2019), the average clause length is 14.77, with a maximum of 4 ECPs per document. Given documents containing d clauses, previous approaches extract ECPs from all possible ($|d| * |d|$) candidate pairs(up to 4 ECPs). Identifying a few valid pairs from numerous candidates is highly challenging. However, if we can preliminarily identify which clauses are more likely to be emotion clauses and filter the relevant cause clauses based on the experiencers in the emotion clauses, we can narrow down the potential ECPs more effectively. This means we can get the true ECPs with a higher probability, therefore, the complexity is reduced compared to $|d| * |d|$ of the search space.

Nevertheless, if the initial emotion clause detection is wrong, it may lead to cascading errors in subsequent steps. To mitigate this issue, we incorporate word-level sentiment domain knowledge into the emotion extraction module to compensate for the shortcomings of emotion instruction prompts. Additionally, introducing an emotional score threshold to filter and select clauses with strong emotional expressions can improve the accuracy of the emotion extraction module. Moreover, the experiencer serves as a crucial link between emotion clauses and cause clauses, guiding the identification of relevant contexts. By fully leveraging experiencer information, we can more accurately correlate emotions with their causes, thereby enhancing the model's overall performance.

### 3.2 TASK DEFINITION

For the ECPE task, the input is a document $D = \{c_1, c_2, \cdots, c_{|d|}\}$, which contains multiple clauses, each clause $ci = \{w_1, w_2, \cdots, w_n\}$ consists $n$ words. The output is all potential emotion-cause pairs (ECPs) $P = \{\cdots, (c_e, c_c), \cdots\}$, where $c_e$ is an emotion clause and $c_c$ represents the corresponding cause clause. It is worth noting that each document contains at least one ECP, and each emotion clause can have multiple cause clauses.

### 3.3 EEEC ARCHITECTURE

We designed an emotion-experiencer-event-cause framework for the ECPE task, a multi-step chain-of-thinking reasoning framework guided by emotion domain knowledge. We aim to leverage LLM's comprehension ability without increasing the training cost and optimize LLM's output with elaborate prompt steps. Figure 2 illustrates the overall architecture. EEEC consists of three key phases: Knowledge-guided Emotion Extraction, experiencer and event extraction, and cause extraction.

### 3.4 KNOWLEDGE-GUIDED EMOTION EXTRACTION

Previous research primarily focused on learning clause embeddings and often ignored the importance of specific words in emotional expression. People typically use specific words to express emotions, providing a direct and effective shortcut to extract emotion clauses. Therefore, we propose identifying emotion clauses by determining specific emotion words in the clauses. Specifically,

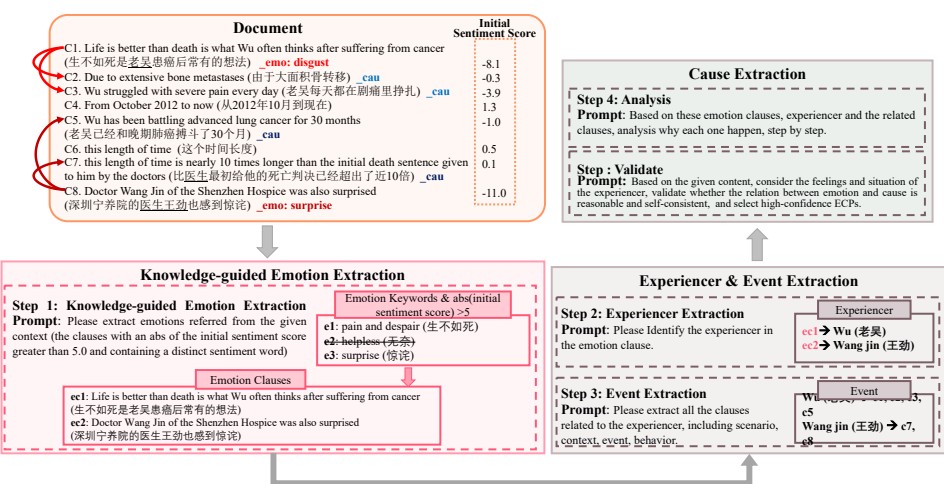

Figure 2: Overview of the proposed EEEC architecture. Experiencers are underlined and highlighted, and different colors are used to distinguish different experiencers; _emotion and _cause refer to the emotion and cause clauses, respectively, and the arrows indicate the cause clauses corresponding to the emotion. In addition, the green highlighted parts represent clauses related to the experiencer.

we design the knowledge-guided emotion extraction module, which uses the clauses's initial sentiment scores computed by the sentiment analysis tool as a priori knowledge to guide the LLM to identify all potential emotion clauses with explicit sentiment expressions and keywords.

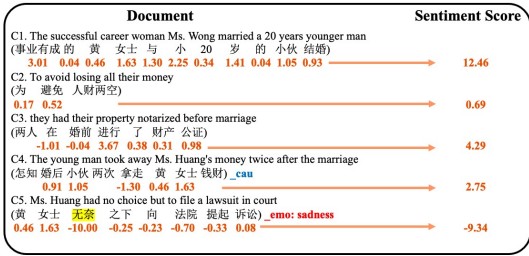

Figure 3: Example for sentiment score learning. c5 is an emotional clause and c4 is a causal clause.

### 3.4.1 SENTIMENT SCORE LEARNING

For the benchmark datasets in the ECPE task, we have observed that emotion clauses usually contain specific clue words that clearly express emotions. As illustrated in Figure 3, C5 represents the emotion clause, C4 denotes the corresponding cause clause, and the other clauses are neither emotion nor cause clause. The yellow-highlighted words primarily determine the clause type, and they convey a more pronounced emotional tone than other terms. To quantify the emotional intensity of each clause, we introduced the Pysenti[2] to calculate the sentiment score for each clause.

Pysenti is a rule-based sentiment polarity analysis method that integrates several sentiment lexicons, including HowNet, the Tsinghua University Li Jun sentiment lexicon, the BosonNLP[3], and a lexicon of negation words. It also provide sentiment scores for common words. In calculating the sentiment score of a clause, $s_i^j$ denotes the $j$-th word's sentiment score of clause $i$. The sentiment score of the entire clause $i$-th is the sum of the sentiment scores of all the words in the clause, as

---

[2]Chinese Sentiment Classification Tool for Python: https://github.com/shibing624/pysenti/blob/master/README.md
[3]http://static.bosonnlp.com/dev/resource

shown in the formula:

$$s_i = \sum_{j=1}^{n} s_j^i. \tag{1}$$

It is worth noting that relying solely on specific sentiment lexicons for emotion polarity analysis can lead to biases. As shown in Figure 3, C1 is not an emotion clause, but it contains many positive words such as "successful career," "small," and "married," which result in an overall higher sentiment score for the clause. In contrast, the actual emotion clause, C5, contains the emotional word "helpless," which has a higher sentiment score. However, the other words in the clause have more dispersed scores, failing to form a similarly emotional solid polarity. To address this, we aim to enhance the sentiment score of emotion clauses by strengthening the use of domain-specific emotional keywords for more accurate sentiment analysis. We collected sentiment keywords from the existing benchmark dataset and assigned values of 10 and $-10$ according to their positive or negative polarity.

In the case of Figure 3, we segment the input text by the Pysenti, then traverse the sentiment lexicon to identify the sentiment polarity of each word, including positive, neutral, and negative sentiments. Pysenti also considers the sentence structure (e.g., conjunctions, negatives, adverbs, punctuation, etc.), assigning weights to the sentiment polarity of each word and then finally obtaining the clause's sentiment score by weighted summation.

### 3.4.2 EMOTION RECOGNIZING AND FILTERING

**Step 1. Emotion Recognizing and Filtering**: For emotion extraction, we adopt a step-by-step filtering strategy that combines the clause's initial sentiment score with keyword analysis to identify emotional clauses. The Algorithm 1 are as follows:

---

**Algorithm 1** Emotion Clause Recognizing and Filtering

**Input**: $D = \{c_1, c_2, \cdots, c_{|d|}\}$, clause's initial sentiment score
**Output**: Set $KC$, Set $EC$
1: Initialize emotion clause set $EC = \emptyset$.
2: Initialize emotion clause set $KC = \emptyset$.
3: **for** ecah clause $c_i \in D$ **do**
4:     Get initial sentiment score $S(c_i)$ from **Input**.
5:     **if** $|S(c_i)| > emo_{threshold}$ **then**
6:         Identify prominent emotional keywords $K(c_i)$ using LLM.
7:         **if** $KC \neq \emptyset$ **then**
8:             Mark $c_i$ as an emotion clause.
9:             Add $c_i$ to set $EC$.
10:            Add $K(c_i)$ to set $KC$.
11:        **end if**
12:    **end if**
13: **end for**
14: **return** Set $KC$, Set $EC$

---

We set an emotional score threshold to filter the clauses with strong emotional expressions. The threshold helps the model focus on sentences with higher sentiment intensity, avoiding the interference of weak or ambiguous emotions and ensuring that the extracted emotion clauses are more salient and representative. This reduces noise and provides more reliable emotional cues for subsequent analysis steps.

### 3.5 EXPERIENCER AND EVENT EXTRACTION

The experiencer is the direct expresser or recipient of the emotion, and the emotion and cause clause for the same sentiment usually refer to the same experiencer. The cause clause typically describes the background or event. Accurate identification of the experiencer and related events is crucial for subsequent cause extraction. Determining the experiencer and events can significantly narrow the search space for cause clauses.

**Step 2. Experiencer Extraction**: Given potential emotion clauses, the LLM is prompted to analyze the experiencer of each sentiment clause and classify each emotion clause according to the following emotion categories: $E = \{$ Happiness, Sadness, Anger, Disgust, Surprise, Fear$\}$.

**Step 3. Event Extraction**: Based on the identified experiencers, consider the interactions between different experiencers and prompt the LLM analyze and summarize the context, background, events and behaviors clauses associated with them in a step-by-step manner.

### 3.6 CAUSE EXTRACTION

Identifying cause clauses directly from the above results may lead to disastrous results, given the complexity of the ECPE task and the uncontrollability of the LLM model. Additionally, every part of the reasoning process could be considered a cause. At the same time, only a few are actual cause clauses, making it relatively difficult to obtain the final ECPs directly. Furthermore, ensuring consistency in the results is challenging due to the unpredictability of zero-shot outputs. To address these issues, we have designed the following two steps:

**Step 4. Analysis**: In this step, based on the above analysis, consider the chronological order and logical correlation between clauses, the cause that directly or indirectly lead to the emotion clauses are analyzed step by step, making sure that the cause clauses are directly related to the emotion clauses and do not contain additional speculative content. It is worth noting that the emotion clauses themselves may also be cause clauses.

**Step 5. Validate**: At this stage, we combine the results from previous multi-step reasoning with a comprehensive understanding of the document to verify whether the relationship between emotion and cause clauses is reasonable and coherent. Specifically, Based on the multiple potential cause clauses found in Step 4, the most direct causes are prioritized, excluding speculative clauses or providing only background or contextual information. Finally, these high-confidence emotion-cause pairs are selected and output in the format [emotion clause number, cause clause number].

## 4 EXPERIMENT

### 4.1 DATASET AND EVALUATION METRICS

We evaluate our approach on three public datasets: (1) The Chinese benchmark dataset published by Ding et al. (2019) consists of Sina City News. (2) The English dataset NTCIR-13 workshop Singh et al. (2021) consists mainly of English novels. (3) The rebalanced Chinese ECPE dataset Ding & Kejriwal (2020) is constructed by resampling Chinese benchmark dataset. In addition, Our approach does not rely on training data, so we used the same dataset split as previous works and performed experiments directly on the test set.

Similar to previous work, we evaluate the ECPE task using precision (P), recall (R), and F1 score as performance metrics. However, assessing ECPE tasks can be challenging due to the inherent uncertainty associated with generative models Han et al. (2023). LLMs typically generate reasonable responses that are correct with human review but may not match the ground truth word-for-word Wadhwa et al. (2023). For example, if the factual cause sentence is "My money was stolen by a thief," an LLM might generate "The thief took the author's money, which likely triggered the author's emotions." While the semantic content of the LLM's output matches the target, the literal wording may differ, and such output should be considered correct. For this, we also used the manual evaluation designed Wang et al. (2023).

### 4.2 BASELINES

We select the following representative methods for comparison, covering approaches from traditional 2-Step frameworks to the latest prompt learning and large language model (LLM)-based methods. ECPE-2Steps: the initial two-step framework. EDSECPE Li et al. (2024): a select-then-extract learning framework for ECPE. RL-TSM Chen et al. (2023): fusion reinforcement learning two-stage ECPE approach. MV-SHIF Yang et al. (2024): treats the ECPE task as a textual entailment problem, modelling the relationship between emotion and cause through inference. EoCP Hu et al. (2024): uses a closed-loop structure detection approach to redefine the structure of ECPs. ECPE-2D Ding

Table 1: The main results compare our EEEC model with the existing three benchmark dataset benchmark methods. The best results are marked in bold.

| Approach | Chinese Dataset | | | English Dataset | | | Rebalanced CN Dataset | | |
|---|---|---|---|---|---|---|---|---|---|
| | P (%) | R (%) | F1 (%) | P (%) | R (%) | F1 (%) | P (%) | R (%) | F1 (%) |
| ECPE-2Step Ding et al. (2019) | 67.21 | 57.05 | 61.28 | 46.94 | 41.02 | 43.67 | | | |
| EDSECPE Li et al. (2024) | 75.60 | 67.18 | 71.14 | - | - | - | - | - | - |
| RL-TSM Chen et al. (2023) | 76.04 | 75.84 | 75.90 | - | - | - | - | - | - |
| MV-SHIF Yang et al. (2024) | 85.00 | 80.70 | 82.80 | 68.40 | 67.30 | 67.90 | | | |
| EoCP Hu et al. (2024) | 79.20 | 76.94 | 78.42 | - | - | - | - | - | - |
| ECPE-2D Ding et al. (2020a) | 72.92 | 65.44 | 68.89 | 60.49 | 43.84 | 50.73 | 47.22 | 37.38 | 41.73 |
| IA-ECPE Huang et al. (2023) | 69.80 | 60.56 | 64.78 | 60.14 | 43.03 | 50.05 | | | |
| E2EECPE Song et al. (2020) | 64.78 | 61.05 | 62.80 | 50.02 | 37.16 | 42.63 | | | |
| MTST-ECPE Fan et al. (2021) | 77.46 | 71.99 | 74.63 | 52.37 | 43.54 | 47.47 | 51.99 | 40.34 | 44.93 |
| UTOS Cheng et al. (2021) | 73.89 | 70.62 | 72.03 | 55.69 | 48.03 | 51.53 | 42.76 | 28.95 | 34.14 |
| ECPE-MLL Ding et al. (2020b) | 77.00 | 72.35 | 74.52 | 52.96 | 45.30 | 51.21 | **61.53** | 36.39 | 45.57 |
| BMST Liu et al. (2023) | 78.01 | 75.49 | 76.67 | - | - | - | - | - | - |
| PairGCN Chen et al. (2020b) | 67.42 | 76.65 | 71.64 | 41.51 | 68.53 | 51.63 | | | |
| RankCP Wei et al. (2020) | 71.19 | 76.30 | 73.60 | 44.00 | 45.35 | 44.63 | 43.22 | 39.16 | 41.22 |
| MGSAG Bao et al. (2022a) | 77.43 | 73.21 | 75.21 | - | - | - | - | - | - |
| EDKA-GM Li et al. (2023b) | 79.1 | 76.08 | 77.56 | - | - | - | - | - | - |
| MGGA Chen & Mao (2023) | 72.95 | 71.02 | 71.89 | 60.56 | 48.95 | 54.04 | | | |
| GAT-ECPE Zhu et al. (2024) | 72.65 | 77.52 | 74.92 | - | - | - | - | - | - |
| A2Net Chen et al. (2022b) | 75.03 | 77.80 | 76.34 | - | - | - | - | - | - |
| JCB Feng et al. (2023) | 79.10 | 75.84 | 77.37 | - | - | - | - | - | - |
| MMN Shang et al. (2023) | 76.11 | 73.96 | 75.02 | - | - | - | - | - | - |
| MM-ECPE Fu & Li (2024) | 77.48 | 76.35 | 76.91 | - | - | - | - | - | - |
| KMGP Zong et al. (2024) | 74.25 | 62.08 | 67.30 | - | - | - | - | - | - |
| DQAN Sun et al. (2021) | 77.32 | 63.70 | 69.79 | **80.58** | 43.24 | 56.28 | | | |
| MM-R Zhou et al. (2022) | 82.18 | 79.27 | 80.62 | 60.55 | 46.88 | 52.08 | | | |
| CD-MRC Cheng et al. (2023) | 82.49 | 78.00 | 80.13 | 60.65 | 46.21 | 52.43 | | | |
| CFC-ECPE Mai et al. (2024) | 82.49 | 81.25 | 81.87 | 61.44 | 53.12 | 56.85 | | | |
| UECA-Prompt Zheng et al. (2022) | 71.82 | 77.99 | 74.70 | - | - | - | 46.30 | 53.22 | 49.37 |
| EmoPrompt-ECPE Gu et al. (2024) | **93.15** | **92.19** | **92.39** | 79.71 | **80.02** | 79.20 | - | - | - |
| GPT3.5 prompt(0-shot) Wang et al. (2023) | 40.74 | 67.54 | 50.82 | 42.11 | 39.34 | 40.68 | 40.72 | 38.14 | 39.39 |
| GPT3.5 DECC(0-shot) Wu et al. (2024) | 61.54 | 49.76 | 55.03 | 34.60 | 59.84 | 43.84 | 45.82 | 49.15 | 47.42 |
| GPT3.5 DECC(4-shot) | 61.23 | 81.56 | 69.95 | 46.89 | 54.42 | 50.35 | 50.00 | **79.45** | 61.38 |
| GPT-4o mini EEEC(0-shot) | 66.74 | 77.63 | 71.75 | 43.15 | 63.96 | 58.42 | 60.53 | 73.32 | **66.31** |

et al. (2020a), EPO-ECPE Hu et al. (2023), IA-ECPE Huang et al. (2023), and E2EECPE Song et al. (2020) transform the ECPE task into a link prediction task. MTST-ECPE Fan et al. (2021), UTOS Cheng et al. (2021), BMST Liu et al. (2023), convert the ECPE task into sequence labelling or multi-label classification problems. Graph-based methods have become another important direction of development. PairGCN Chen et al. (2020b), RankCP Wei et al. (2020), MGSAG Bao et al. (2022a), EDKA-GM Li et al. (2023b), MGGA Chen & Mao (2023), GAT-ECPE Zhu et al. (2024), and other graph-based methods. A2Net Chen et al. (2022b), RSN Chen et al. (2022a), JCB Feng et al. (2023), MMN Shang et al. (2023), MM-ECPE Fu & Li (2024), KMGP Zong et al. (2024) and other inter-task interaction-based learning methods. Question-answering (QA) or machine reading comprehension (MRC)-based models, like DQAN Sun et al. (2021), Guided-QA Nguyen & Nguyen (2023), MM-R Zhou et al. (2022), CD-MRC Cheng et al. (2023), and CFC-ECPE Mai et al. (2024). UECA-Prompt Zheng et al. (2022) and EmpPrompt-ECPE Gu et al. (2024) introduce prompt-based learning, guiding the model to extract and pair emotion components with specific prompts. Reasoning methods, such as DECC Wu et al. (2024), GPT3.5-promptWang et al. (2023), use ChatGPT as a baseline framework.

## 4.3 RESULTS AND ANALYSIS

### 4.3.1 ZERO-SHOT RESULT

In the experiments, we used GPT-4o mini as the baseline model to evaluate the proposed EEEC across three benchmark datasets: Chinese Ding et al. (2019), English Singh et al. (2021), and the balanced Ding & Kejriwal (2020) dataset. The results are shown in Table 1. For the Chinese dataset, our EEEC framework outperformed existing large language model (LLM) methods, achieving the best results. Notably, in the zero-shot setting, EEEC demonstrated significant improvements over the four-shot DECC in both P and F1 score, with P increasing by 5.51 and F1 by 1.8. This improvement can be attributed to incorporating prior sentiment knowledge in Step 1, which better guided the model in extracting emotion clauses. Additionally, the experiencer identification in Step

3 helped narrow the search space for cause clauses. Compared to fully-supervised fine-tuning methods, we found that the zero-shot EEEC outperformed the select-then-extract framework EDSECPE, with a $0.61$ improvement in F1. EEEC also showed clear advantages over end-to-end methods like ECPE-3D, IA-ECPE, and E2EECPE, with F1 improvements of $2.86$, $6.97$, and $8.95$, respectively. Compared to the graph-based PairGCN method, EEEC achieved a higher F1 by $0.11$. Despite these strong performances, EEEC still falls short of most fully-supervised fine-tuning methods.

In contrast to its performance on the Chinese dataset, EEEC outperforms most fully-supervised fine-tuning methods on the English benchmark dataset, surpassing even the leading LLM-based DECC model. Specifically, EEEC achieves an F1 score improvement of $8.07$ compared to DECC. This improvement is primarily due to DECC's reliance on emotion keywords when extracting emotion clauses, which inevitably leads to omissions. Furthermore, DECC's approach to extracting cause clauses focuses solely on deriving them from emotion clauses, overlooking the critical relationships between experiencers, emotion clauses, and cause clauses. In contrast, EEEC more effectively captures the role of experiencers and integrates contextual information, resulting in a substantial performance boost.

### 4.3.2 DE-BIAS RESULT

In the Chinese benchmark dataset, 80% of the cause clauses are either emotion clauses themselves or located in adjacent clauses, resulting in a significant positional bias. Therefore, we conducted evaluations on the rebalanced dataset. The results are shown in Table 1, where all previous methods based on fully supervised fine-tuning show a significant performance degradation on the rebalanced dataset. In contrast, EEEC's F1 score decreases significantly less than that of the other models in the zero-shot setting. Furthermore, EEEC in the zero-shot setting outperforms the state-of-the-art methods on this rebalanced dataset.

We analyzed this phenomenon: previous methods often exploit positional bias to extract emotion-cause pairs, which can overfit this bias, resulting in false correlations and neglecting the semantic features of the text, which leads to significant performance degradation. In contrast, the EEEC method relies entirely on the LLM's inherent understanding and reasoning abilities, making it less sensitive to the positional distribution within the dataset. Additionally, we leverage prior sentiment knowledge to guide the LLM, enabling it to identify emotions more accurately.

### 4.3.3 PERFORMANCE ON MULTIPLE PAIRS

In addition, we compared the results of extracting emotion-cause pairs from documents with multiple pairs. Specifically, we divided each test set into two groups: one group containing only a single Emotion-Cause Pair (ECP) and the other group containing two or more ECPs. The comparison results are shown in Table 2. Our model EEEC performs exceptionally well in multi-pair extraction scenarios, achieving an F1 score that surpasses the previous best model DEEC by $4.8$. This improvement indicates that the guidance of prior emotional knowledge and incorporating experiencer features are efficient for complex emotional reasoning. However, it fails to achieve optimal performance in terms of P. This is primarily due to the complex structure and semantic relationships in documents containing multiple pairs. While introducing prior emotional knowledge helps the model extract potential emotion clauses, it inevitably draws attention to the implicit emotion clauses within the document, leading to the extraction of more invalid pairs.

Table 2: Results on multi-pair extraction scenarios. The best results are marked in bold.

| Methods | P(%) | R(%) | F1(%) |
|---|---|---|---|
| UTOS | 55.45 | 46.76 | 50.35 |
| RankCP | **75.08** | 43.90 | 55.31 |
| ECPE-MLL | 70.45 | 47.76 | 56.88 |
| UECA-Prompt | 69.52 | 54.66 | 61.14 |
| UECA-Prompt (m2m) | 73.92 | 56.30 | 63.45 |
| GPT3.5 DECC (4-shot) | 63.93 | **73.39** | 68.34 |
| GPT4o mini EEEC | 73.57 | 72.83 | **73.14** |

## 5    ABLATION STUDY

### 5.1    DIFFERENT COMPONENTS

To evaluate each step's impact on the EEEC model's performance, we conducted experiments on variations of the EEEC where each step was removed. Specifically, there are three variations related to the removal of Step 1: w/o step1-para (this variation ignores the prior emotional knowledge of the clauses), w/o step1-keyword (this variation does not treat emotional keywords in the clauses as critical features), and w/o step1 (this variation directly identifies emotion clauses from the document). Table 3 shows the experimental results. When prior emotional knowledge is ignored, the F1 score drops more significantly than when emotional keywords are disregarded, indicating that emotional knowledge helps the LLM explore the possibility of clauses being emotion clauses. Removing the emotion clause identification step leads to a significant performance drop, as providing excessively noisy clauses to subsequent steps without any constraints introduces confusion. Removing the experiencer identification step results in unrelated clauses being considered candidate cause clauses, which may lead to inaccurate cause analysis and reduced precision. On the other hand, removing the step of identifying experiencer-related contexts, backgrounds, events, and behaviour clauses has a minor impact on performance. This step primarily helps summarize the experiencer's story to aid the model in understanding the document's main content in a more targeted manner. Eliminating the cause analysis step sharply declines performance. An end-to-end approach to extracting cause clauses directly from the document overlooks the LLM's reasoning capabilities. However, removing the verification and summary steps has minimal impact on performance.

Table 3: Ablation study on Chinese dataset.

| Methods | P(%) | R(%) | F1(%) |
|---|---|---|---|
| EEEC | 66.74 | 77.63 | 71.75 |
| w/o step1-para | 38.38 | 58.24 | 45.91 |
| w/o step1-keyword | 41.53 | 53.84 | 46.87 |
| w/o step1-keyword&para | 34.89 | 49.18 | 40.74 |
| w/o step2-experiencer | 57.38 | 62.32 | 59.21 |
| w/o step3-event | 59.29 | 64.84 | 62.11 |
| w/o step4-analysis | 61.82 | 57.46 | 58.63 |
| w/o step5-validate | 65.87 | 78.24 | 70.64 |

## 6    CONCLUSION

Considering that previous works ignore the influence of experiencer, a priori emotion knowledge on emotion-cause on extraction task (ECPE), and the problem of spurious correlations caused by positional bias overfitting, this paper proposes the Emotion-Experience-Event-Cause multi-step chain (EEEC) framework. By decomposing the ECPE task into multiple sub-tasks, with each solution depending on the previous step, EEEC leverages the reasoning capabilities of large language models (LLMs) through well-designed prompts and chain-of-thought decomposition, addressing the limitations of fully supervised fine-tuning methods. To obtain experiencer information, we introduced a sub-task for experiencer identification, capturing the relationship between experiencers, emotion clauses, and cause clauses. Furthermore, we designed sub-tasks to extract experiencer-related contexts and events, reducing the search space for cause clauses. To accurately extract emotion clauses, the framework introduces a priori sentiment knowledge to guide LLM to recognize emotion clauses more effectively. Experiments conducted on three benchmark datasets demonstrate the effectiveness and robustness of the EEEC framework. Especially in complex multi-pair ECPE tasks, EEEC significantly outperforms state-of-the-art methods. More importantly, EEEC maintains high performance in zero-times learning scenarios, demonstrating its strong inference capability and adaptability to different language datasets. Future work will focus on developing more efficient LLM-based models and improving performance by extracting complex emotion-cause relationships.

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

# A APPENDIX

## A.1 DATASET STATISTICS

## A.2 IMPLEMENTATION DETAILS

Consistent with the approach reported by Wang et al. (2023) and Wu et al. (2024), we generated all ChatGPT outputs using the official API, with default hyperparameters such as temperature and top-k sampling. The experimental results represent the average of five random runs on the test set. Additionally, we conducted a manual evaluation involving five annotators. For each instance, the result was correct only when three or more annotators agreed that the LLM-generated answer was correct.

Table 4: Here are the statistics for the three benchmark dataset used for Emotion-Cause Pair Extraction.

| Statistics | Chinese Dataset | | English Dataset | | Rebalanced Dataset | |
|---|---|---|---|---|---|---|
| Range | Number | Percentage (%) | Number | Percentage (%) | Number | Percentage (%) |
| Documents | 1945 | 100 | 2843 | 100 | 756 | 100 |
| One pair in documents | 1746 | 89.77 | 2537 | 89.24 | 715 | 94.58 |
| Two pairs in documents | 177 | 9.10 | 256 | 9.00 | 40 | 5.29 |
| More than two pairs | 22 | 1.13 | 50 | 1.76 | 1 | 0.17 |
| Total pairs | 2167 | 100 | 3215 | 100 | 780 | 100 |
| distance is equal to 0 | 511 | 23.6 | 1640 | 51.01 | 169 | 21.67 |
| distance is equal to 1 | 1342 | 61.9 | 825 | 25.66 | 331 | 42.44 |
| distance is equal to 2 | 224 | 10.3 | 328 | 10.21 | 196 | 25.13 |
| distance is equal to 3 | 50 | 2.3 | 156 | 4.85 | 43 | 5.51 |
| distance is large than 3 | 40 | 1.9 | 266 | 8.27 | 41 | 5.25 |

## A.3 CASE STUDY

In some cases, the EEEC framework is erroneous, mainly in the following situations: One: Extracting implicit sentiment expressions in documents. As shown in Figure 4, we can find from the analysis process given by LLM that leaving is not a typical emotion word when performing emotion word verification in step 2, but judging from the context, clause 6 implies the emotion of sadness. Two: Tendency to extract deeper emotional causes. As seen in Figure 5, during the cause extraction process, EEEC goes beyond extracting superficial information such as "Deng Guangtai's story" and instead identifies the more profound emotional significance behind the story—"family affection"—demonstrating its strength in emotional reasoning.

**Document:**
1,1月20日上午,1.0
2,望江县华阳镇一所学校,0.3
3,学生们从老师手里接过成绩报告册,1.0
4,愉快的寒假正式开始,12.9
5,然而就在当晚,-1.3
6,9岁的三年级学生小林化名却永远地离开了这个世界,5.6
7,孩子被发现时,-1.5
8,缢死在厕所的横梁上,-3.1
9,警方初步认定是自缢身亡,1.5
10,学校调查称,-0.6
11,当天吃晚饭时,1.4
12,孩子外公外婆说爸妈不回来过年,0.5
13,小林的情绪很低落,-9.0
**Ground Truth:**
情感子句：小林的情绪很低落
原因子句：孩子外公外婆说爸妈不回来过年
情感原因对：(13, 12)

**LLM Output:**
### Step 1: 找出初始情感得分绝对值大于5.0且包含明确情绪词的子句

6,9岁的三年级学生小林化名却永远地离开了这个世界,5.6
13,小林的情绪很低落,-9.0

### Step 2: 验证情绪词并分类情感子句

情绪词验证：
- **"离开"**：虽然这不是一个典型的情绪词，但是结合语境，"永远地离开了这个世界"显示了深深的悲伤和失落。
- **"情绪很低落"**：这是一个明确的情绪表达词。

情感体验者与分类：
1. **子句6**：情感体验者是小林（化名），情感类别为悲伤。
2. **子句13**：情感体验者是小林（化名），情感类别为悲伤。

情感子句：
- 子句6: 9岁的三年级学生小林化名却永远地离开了这个世界
  情感体验者：小林
  情感类别：悲伤

- 子句13: 小林的情绪很低落
  情感体验者：小林
  情感类别：悲伤

### Step 3: 总结情境、背景、事件、行为子句
时间、地点和角色背景描述：
- **时间**：1月20日上午，愉快的寒假正式开始（子句1和子句4）。
- **地点**：望江县华阳镇一所学校（子句2）。
- **角色背景**：9岁的三年级学生小林（子句6）。

相关子句：
- **子句1**：1月20日上午。
- **子句2**：望江县华阳镇一所学校。
- **子句3**：学生们从老师手里接过成绩报告册。
- **子句4**：愉快的寒假正式开始。
- **子句5**：然而就在当晚。
- **子句6**：9岁的三年级学生小林化名却永远地离开了这个世界。
- **子句7**：孩子被发现时。
- **子句8**：缢死在厕所的横梁上。
- **子句9**：警方初步认定是自缢身亡。
- **子句10**：学校调查称。
- **子句11**：当天吃晚饭时。
- **子句12**：孩子外公外婆说爸妈不回来过年。
- **子句13**：小林的情绪很低落。

角色之间的联系和交互：
- 小林与家长（外公外婆）之间存在互动（子句12）。
- 小林与学校和老师之间的联系（子句2, 3, 4, 6, 10）。
### Step 4: 找出导致情感子句产生的原因子句
- 情感子句6:
- **子句8**：缢死在厕所的横梁上。
- **子句9**：警方初步认定是自缢身亡。
- 情感子句13:
- **子句12**：孩子外公外婆说爸妈不回来过年。
### Step 5: 优先选择最直接的原因子句
- [13, 12]

Figure 4: The case of extracting implicit sentiment expressions in documents.

**Document:**
1,痴汉邓广台的故事感动了许多人,12.0
2,很多人都说,-0.9
3,他,0
4,让人深切地感受到了亲情的含义,0.3
5,今天或许我们已经离开家在社会奋斗,1.1
6,或正要离开家,0.5
7,但是我们之所以这么勇敢地面对有时显得残忍社会,1.0
8,就是因为我们相信,2.4
9,家人永远是自己的陪伴,4.6

**Ground Truth:**
情感子句: 痴汉邓广台的故事感动了许多人
原因子句: 痴汉邓广台的故事感动了许多人
情感原因对：(1, 1)

**LLM Output:**
### Step 1: 找出初始情感得分绝对值大于5.0且包含明确情绪词的子句
我们先检查每个子句的初始情感得分，找出绝对值大于5.0的子句，并寻找明确的情绪词。
子句1的初始情感得分为12.0，绝对值大于5.0。我们检查是否包含明确的情绪词：
- 子句1包含"感动"这一明确的情绪词。
所以，找到的情感子句是子句1，情绪词是"感动"。

### Step 2: 验证情绪词并分类情感子句
验证"感动"是否为情绪词：
- "感动"是一个情绪词，表达一种情感体验。

找出情感体验者：
- 子句1的情感体验者是"许多人"。

将情感子句按情感类别分类：
- "感动"属于"高兴"类别。

情感子句、情感体验者和情感类别如下：
- 子句1: 痴汉邓广台的故事感动了许多人, 情感体验者: 许多人, 情感类别: 高兴

### Step 3: 总结情境、背景、事件、行为子句及交互
结合情境信息，描述与情感体验者相关的背景和事件：
- 时间：未明确提及
- 地点：未明确提及
- 角色背景：邓广台的故事
- 事件：邓广台的故事感动了许多人
情感体验者"许多人"与邓广台之间的主要联系是邓广台的故事使"许多人"感动。

### Step 4: 找出导致情感子句的原因子句
找出直接或间接导致情感子句（子句1）产生的原因子句：
- 子句1的情感体验是"许多人感动"，我们需要找出为什么许多人感动的原因子句。

考虑到时间顺序和逻辑关联，以下子句可能是导致情感子句的原因：
- 子句4: 让人深切地感受到了亲情的含义,0.3
- 子句9: 家人永远是自己的陪伴,4.6

### Step 5: 选择最直接的原因子句
在子句4和子句9中，选择最直接的原因子句：
- 子句4（让人深切地感受到了亲情的含义）是导致"感动"的直接原因，因为它描述了故事带来的情感共鸣
所以，最直接的原因子句是子句4。
最终输出格式如下：
- [1, 4] (痴汉邓广台的故事感动了许多人是因为让人深切地感受到了亲情的含义)

总结：
1. 情感子句: 痴汉邓广台的故事感动了许多人
2. 原因子句: 让人深切地感受到了亲情的含义

Figure 5: The case of extracting deeper emotional causes.

