# OpenReview forum: "EEEC: Emotion-Experiencer-Event-Cause multi-step chain reasoning for Emotion-Cause Pair Extraction"
_ICLR.cc/2025/Conference — ICLR 2025 Conference Withdrawn Submission_

### Official Review · Reviewer_483Q · 2024-10-17

**Soundness:** 2
**Presentation:** 2
**Contribution:** 1
**Rating:** 3
**Confidence:** 4

**Summary:**

This paper proposes the Emotion-Experience-Event-Cause (EEEC) framework for the emotion-cause pair extraction (ECPE) task. Specifically, the EEEC framework includes three steps: Knowledge-guided Emotion Extraction, Experiencer & Event Extraction, and Cause Extraction. Experimental results show its effectiveness under the zero-shot setting.

**Strengths:**

1. The overall framework is clear.
2. The experimental results are good.

**Weaknesses:**

1. The presentation of some details is poor. (1) Please consider using the same documents in Figures 1 and 2 to aid understanding. (2) I suggest polishing Experiencer & Event Extraction and Cause Extraction in Figure 2. (3) KC in algorithm 1 seems to be key words set.
2. This paper uses a different base compared to GPT3.5 prompt and GPT3.5 DECC, which may affect the evaluation.
3. Where is w/o step1 in Table 3? Will directly extracting emotion clauses using LLMs result in poor performance in extracting emotion clauses?
4. The novelty and transferability of this paper appear to be average. For example, experiencer and event extraction, and analysis and validate.

**Questions:**

1. On which dataset are the results in Table 2? Does this indicate that the proposed method performs better on many pairs of documents and worse on a single pair?
2. Why does the proposed method perform better on the English dataset than the Chinese dataset compared to other baselines?
3. Please consider changing the form of the reference in Section baselines.

---

### Official Review · Reviewer_eXEt · 2024-11-03

**Soundness:** 2
**Presentation:** 2
**Contribution:** 2
**Rating:** 3
**Confidence:** 5

**Summary:**

The paper introduces the Emotion-Experiencer-Event-Cause (EEEC) framework, which improves the accuracy of finding emotion-cause pairs in text. Emotion-Cause Pair Extraction (ECPE) involves identifying emotions and their causes in text, useful for sentiment analysis. Traditional methods have issues with bias and lack of specialized knowledge. EEEC tackles this by breaking down the task into steps, with a focus on identifying “experiencers” (who feels the emotion) to better pinpoint causes. Using large language models like GPT-4, the framework shows improved performance in both English and Chinese without additional training.

**Strengths:**

Strengths:
1. The paper is well-organized and easy to read.
2. By using LLMs, EEEC achieves promising results without fine-tuning, which suggests adaptability across languages and datasets.

**Weaknesses:**

Weaknesses:
1.	 The framework’s reliance on LLMs may limit its application in scenarios where computational resources are restricted.
2.	The multi-step approach, while effective, introduces potential challenges in managing and calibrating each sub-task, which could complicate implementation in practical scenarios.
3.	Errors in initial steps, especially emotion clause identification, can propagate through subsequent stages, impacting final accuracy. Although the authors attempted to mitigate the error propagation, the multi-step method may inevitably suffer from this issue, especially in processing hard samples.
4.	EEEC may struggle with implicit emotion expressions or subtle emotional nuances, as noted in some case studies.
5.	Although zero-shot results are promising, performance still lags behind fine-tuned models on specific datasets, suggesting that supervised approaches retain an edge in precision.
6. The presentation of table 1 and the introduction of baselines are not well-organized. There is no need to include numerous baselines (SOTA and representative methods are enough) )as you cannot analyze all the baselines.
7. The format of tables and figures are not consistent across the paper, e.g., table 1.

**Questions:**

NA

---

### Official Review · Reviewer_raHV · 2024-11-04

**Soundness:** 2
**Presentation:** 2
**Contribution:** 2
**Rating:** 3
**Confidence:** 4

**Summary:**

## Summary:
This paper introduces a multi-step reasoning approach for the problem of Emotion Cause Pair Extraction (ECPE). It breaks the problem down into a five step process as part of the proposed Emotion-Experience-Event-Cause (EEEC) framework: 1/ identifying emotion clauses guided by word level scores passed to the LLM, 2/ extracting experiencers for these emotions, 3/ extracting events to provide the right context 4/ clause extraction using step-by-step LLM chain-of-thought and reflection where the LLM itself validates the final results.

For 1, the paper uses a rule-based sentiment polarity analysis method that gives sentiment scores to each word. These word level scores are aggregated per sentence for the final emotion scores. These are passed to the LLM in Step 1, in addition to keyword detection and scoring. For 2 and 3, they prompt the LLM to detect the spans in the text that are experiencers and events respectively for the emotion clauses found in step 1. Next, in step 4, the LLM is asked to use everything it has found so far and analyze each clause for being a valid cause. Finally, in step 5, the LLM is asked to reflect on its own result so far and validate if it is coherent and self-consistent.

The paper presents a comparison of the approach with a long list of other models on three datasets, one of which is notably rebalanced to avoid overindexing on the positional closeness between emotion and cause. Additionally, it also presents an ablation study that ranks the value of each of the 5 steps above. EEEC improves over the SoTA prominently in the zero-shot setting and when there are more than one pairs of ECs in the documents.

## Overall Recommendation:
The paper, as it stands here, should be rejected because (1) the soundness of the paper in how it does step 1, how it compares to other approaches, how it reindexes the test set and why it doesn't report numbers of other comprehensive competitors on the reindexed benchmark is unclear making the final outcomes less strong (2) the writing is unnecessarily complex and has many unsubstantiated claims (3) reproducibility is unclear without a clear description of prompts (4) some methods like word level sentiment score aggregation don't have simple and most intuitive baselines such as sentence level sentiment analysis.

**Strengths:**

The paper approaches the ECPE problem with a deep understanding of the nuances of the task which enables the authors to use various task specific heuristics for solving subproblems.
1/ Originality: The paper's main novelty comes through as the combination of the 5 steps and chaining them in LLMs though each of them individually has been explored in the past. The paper presents a de-biased dataset to overcome the bias towards the positional closeness of an emotion and its cause which is a dataset artifact.
2/ Significance: The results indicate that the approach especially stands out when dealing with multiple pairs of ECs in the task and similarly in the zero-shot setting.

**Weaknesses:**

## Methodology
1. The method for sentiment based word level score aggregation is not sufficiently motivated. Specifically, (a) not normalizing the score penalizes short sentences, (b) sentences with negations may not be handled since scores are word level without using sentence level context and semantics (c) there is no comparison of this rule-based word level scoring with an LLM's inherent ability to assess sentiment. For example, a simple comparison would be to instruct the LLM to give sentence level scores/ranks based on overall sentiment as an explicit step and use that instead of this word level scoring.
2. It is unclear how L367-369 would work. If the emotion clauses finally generated by the LLM are not verbatim but still marked correct, the task as defined in 3.2 does not hold.
3. Table 1 shows the more "trustworthy" dataset titled 'Rebalanced CN dataset' but many comparison models don't have that filled in. The other columns seem self-reported?


## Unsubstantiated claims:
1. L83-84 claim that using any of the extra information requires large amounts of labelled training data. This is not true, since you could use any of this information by weak labeling using one of these models or prompting the LLM to specifically consider these factors and even make these scores explicit in a step-by-step approach.
2. L95-96 is a claim that document level processing will "inevitably consider redundant information". It is unclear what this is based on, specially when LLMs today could be prompted to find relevant information first.
3. L429-431: When comparing to DECC, the paper says that the improvement comes from Step 1. How that isolated conclusion is made is unclear.

## Clarity
1. Prompts are not clearly stated per step, except for Figure 1 (which is a block diagram making the exact prompts hard to infer).
2. Related work and Table 1 have a large list of other models to compare against, but based on Sec 4.3.2 the paper makes a case to evaluate only on their rebalanced dataset. For this, Table 1 does not have results for many of the comparison models. Also, related work talks of a lot of models but does not tease apart the specific differences between theme, draw upon themes across them or draw out the novelty of EEEC over them.
3. L108-109 shows a 5 step process. Then L208-209 talks about 3 key phases. Finally, in section 3.4-3.6 it is again broken down to a 5 step process. Consistency in this will help the readability overall.
4. L339-349 talk about the analyze and validate steps. Based on Figure 1, it seems like this basically means asking the LLM to analyze all inputs and propose an answer. And next, validation seems like an LLM reflection process to confirm final answers. Clarify the working of these steps in the actual phase descriptions.

**Questions:**

1. Why do you consider it a novelty of this paper to give importance to the experiencer? Per your own literature survey too, in L160-161, Lee et al have already proposed experiencer based modeling since 2023. Do you mean that the combination of experiencer with the other components of your pipeline are unique?
2. In step 1, section 3.4.1, L256 onwards, the sentiment score is aggregated over all words without normalizing, is that accurate? If so, wouldn't the longer sentences game the threshold more easily and always be filtered in? What is the reason to not normalize the scores?
3. Per Figure 1, you give the scores to the LLM and ask it to filter the clauses for a threshold? Why is this an LLM call and not a deterministic code piece to filter clauses based on the threshold? Also, how is the threshold of 5.0 decided?
4. Figure 1 shows some prompt structure in english while appendix has others in just Chinese. Can you please add detailed and specific English prompts including how the sentiment scores are passed to the LLM.
5. What are the implications of LLM hallucination where the clause changes as shown in L366-369? If a clause is not extracted verbatim, why should it be considered correct? Doesn't that break the task definition itself per Section 3.2?
6. L257-259: Is this observation based on this current benchmarking data? If so, that would be contamination.
7. Why is emotion classification done with experiencer extraction and not with emotion filtering in step1?

**Details Of Ethics Concerns:**

In Figure 1, the paper would benefit from some ethical considerations regarding stereotypical gender roles and objectifying language, such as the use of "sexy" for the wife in the example which is discussing relationship details in a couple. This terminology may risk reinforcing harmful stereotypes and promotes reductive views of gender. Such language can diminish the perceived complexity of emotions in both genders.

I believe this is exaggerated since this is not a native English example and some of the tone may be coming in from the translation. I do not believe this was intentional from the authors.

---

### Official Review · Reviewer_21YB · 2024-11-09

**Soundness:** 2
**Presentation:** 3
**Contribution:** 2
**Rating:** 3
**Confidence:** 4

**Summary:**

The paper introduces the Emotion-Experiencer-Event-Cause (EEEC) framework, a multi-step reasoning approach for emotion-cause pair extraction (ECPE). By incorporating experiencer identification, prior sentiment knowledge, and logical association between emotion and cause clauses, EEEC aims to improve the accuracy and robustness of extracting emotion-cause pairs, particularly in zero-shot scenarios.

**Strengths:**

1. Integrates domain-specific knowledge and experiencer identification for accurate emotion-cause extraction.
2. Achieves strong zero-shot performance, surpassing some supervised methods.

**Weaknesses:**

1. The proposed methods rely heavily on manually designed rule, which limit their effectiveness in emotion-cause pair extraction, a nuanced area of research. The learning methods section lacks inspiration, and overall, it is not very engaging.
2. It is generally known that the meanings of words change according to the context. Therefore, word-level sentiment domain knowledge is not well-suited for emotion-cause pair reasoning tasks, which require a deep understanding of context.
3. The paper focuses only on cause clause detection, without explaining the rationale behind emotion-cause pair extraction, which is more intuitive for human understanding.

**Questions:**

In Figure 3, the authors only highlighted the words in yellow in the Chinese text, overlooking the English part, which impacts user comprehension.

---

### Note · Authors · 2024-11-14

I have read and agree with the venue's withdrawal policy on behalf of myself and my co-authors.